# Type-I CdSe@CdS@ZnS Heterostructured Nanocrystals with Long Fluorescence Lifetime

**DOI:** 10.3390/ma16217007

**Published:** 2023-11-01

**Authors:** Yuzhe Wang, Yueqi Zhong, Jiangzhi Zi, Zichao Lian

**Affiliations:** School of Materials and Chemistry, University of Shanghai for Science and Technology, Shanghai 200093, China; 212142401@st.usst.edu.cn (Y.W.); 203612315@st.usst.edu.cn (Y.Z.); jiangzhizi@usst.edu.cn (J.Z.)

**Keywords:** CdSe quantum dots, charge separation, type-I heterostructured semiconductor, luminescence quantum efficiency and lifetime

## Abstract

Conventional single-component quantum dots (QDs) suffer from low recombination rates of photogenerated electrons and holes, which hinders their ability to meet the requirements for LED and laser applications. Therefore, it is urgent to design multicomponent heterojunction nanocrystals with these properties. Herein, we used CdSe quantum dot nanocrystals as a typical model, which were synthesized by means of a colloidal chemistry method at high temperatures. Then, CdS with a wide band gap was used to encapsulate the CdSe QDs, forming a CdSe@CdS core@shell heterojunction. Finally, the CdSe@CdS core@shell was modified through the growth of the ZnS shell to obtain CdSe@CdS@ZnS heterojunction nanocrystal hybrids. The morphologies, phases, structures and performance characteristics of CdSe@CdS@ZnS were evaluated using various analytical techniques, including transmission electron microscopy, X-ray diffraction, UV-vis absorption spectroscopy, fluorescence spectroscopy and time-resolved transient photoluminescence spectroscopy. The results show that the energy band structure is transformed from type II to type I after the ZnS growth. The photoluminescence lifetime increases from 41.4 ns to 88.8 ns and the photoluminescence quantum efficiency reaches 17.05% compared with that of pristine CdSe QDs. This paper provides a fundamental study and a new route for studying light-emitting devices and biological imaging based on multicomponent QDs.

## 1. Introduction

In recent years, colloidal nanocrystals have aroused extensive attention in the fields of basic research and practical applications due to their excellent optical and photoelectrical properties [1,2,3,4,5]. For example, they have broad application prospects in the fields of fluorescent probes [6], lasers [7], solar cells [8,9], bioimaging [10,11] and luminescent devices [12,13,14]. Cadmium selenide (CdSe) quantum dots (QDs) have been studied extensively because of their quantum size effects and because their luminous wavelength can be extended to the entire visible region. QDs can be synthesized via surface modification of organic molecules not only to improve their luminescent properties and stability but also to obtain a narrow size distribution [15]. However, the presence of many dangling bonds on the surface of the material may lead to defects, which can greatly reduce the stability and luminescent properties of the material [16]. In order to solve this problem, many researchers adopt a method of surface modification of inorganic materials to construct type-I semiconductor band structures by generating a passivation effect of core@shell heterojunction on the surface of materials [17,18]. For instance, Zhu et al. [19] synthesized CdSe/ZnS type I core–shell quantum dots by choosing wide-band-gap ZnS as the shell material and developed a method to optimize the charge separation rate and lifetime by controlling the thickness of the shell material. Toufanian et al. [20] prepared type I InP/ZnSe/ZnS quantum dots and overcame the inherent brightness mismatch seen in QDs through concerted materials design of heterostructured core/shell InP-based QDs. Panda et al. [21] alloyed CdSe/CdS quantum dots at elevated temperatures to prepare high-quality CdZnSe quantum dots and obtained very high environmental stability. Research has also found that heterostructures can effectively resist their own chemical degradation and photo-corrosion, thereby improving their photoluminescent quantum efficiency. However, type-II semiconductor heterostructures can reduce the photoluminescent quantum efficiency of materials by promoting the separation of photo-generated electrons and holes. Therefore, this method is not conducive to enhancing the luminescent properties of materials [22]. However, there are few reports on the construction of multi-component structures from type II to type I to enhance the luminescence properties of materials [23,24].

In this study, we first constructed type-II heterojunction with CdSe quantum dots as the core and CdS as the shell. Then, we used ZnS with a wide band gap for modification and constructed type-I heterojunction, which enhanced the fluorescence quantum efficiency and luminescence lifetime of the material. After CdSe was coated with CdS, the UV-vis absorption spectrum and fluorescence emission spectrum of CdSe@CdS showed a red shift, the fluorescence lifetime increased from 41.4 ns to 59.0 ns, but the fluorescence quantum efficiency decreased from 15.89% to 6.32%. Additionally, the effects of different proportions of ZnS on the surface modification of CdSe@CdS were studied. The UV-vis absorption spectra and fluorescence emission spectra of CdSe@CdS@ZnS showed a red shift after the modification of ZnS. When the optimal ratio of Zn:S = 0.6:1.2, the fluorescence lifetime of the material increased from 59.0 ns to 88.8 ns and the fluorescence quantum efficiency increased from 6.32% to 17.05%, showing good luminescence properties.

## 2. Results and Discussion

### 2.1. Characterization of Synthetic Materials

We synthesized novel CdSe@CdS@ZnS heterojunction nanocrystals (NCs) and studied their luminescent properties and the lifetime of photogenerated electrons and holes. The whole synthetic processes are shown in Figure 1a. First, CdSe quantum dots were synthesized via the thermally injected wet chemical method using CdO as the cadmium source and a selenium precursor as the selenium source [25]. As shown in Figure 1b, the representative transmission electron microscope (TEM) image shows that the particle size of CdSe QDs was 4.3 ± 0.1 nm (Figure 1e). The CdSe@CdS core@shell structure was obtained by slowly injecting a Cd-and-S-mixed precursor solution. As shown in Figure 1c, its particle size distribution was 6.8 ± 0.1 nm (Figure 1f). Compared with CdSe QDs, the size of CdSe@CdS nanocrystals was increased by 2.5 nm. It could be concluded that CdS was coated on the surface of CdSe, corresponding to the thickness of CdS of about 1.25 nm. Then, we used CdSe@CdS as the core, zinc acetate as the zinc source, and a sulfur–trioctylphosphine (S-TOP) solution as the sulfur source, which were slowly injected into the reaction solution to obtain CdSe@CdS@ZnS heterojunction NCs. The TEM image is shown in Figure 1d, and the size distribution shows that its size was 8.8 ± 0.1 nm (Figure 1g) corresponding to the thickness of ZnS of about 1.0 nm. As the shells were loaded, the size of the materials gradually increased. The HRTEM image in Figure 1h shows the high crystallinity of the CdSe@CdS@ZnS, which revealed the existence of the CdSe, CdS and ZnS phases. The different lattice spacings of 0.32 nm, 0.33 nm and 0.31 nm corresponded to wurtzite CdSe (w-CdSe) (101), zinc-blende CdS (w-CdSe) (111) and zinc-blende ZnS (zb-ZnS) (111), respectively. The HRTEM (Figure 1h), high-angle annular dark-field (HAADF) scanning TEM (STEM) and STEM-energy dispersive X-ray spectrometry (EDS) mapping (Figure 1i–n) of CdSe@CdS@ZnS showed that CdSe was coated with CdS and ZnS.

The crystal structures of the nanomaterials were characterized via X-ray diffraction (XRD). As shown in Figure 2, the synthesized CdSe QDs were w-CdSe (Joint Committee on Powder Diffraction Standards (JCPDS) no. 08-0459) [26]. When CdS was coated on CdSe to form a CdSe@CdS core@shell structure, only the diffraction peak of zb-CdS (JCPDS no. 10-0454) showed in the material, which might be due to the thickness of the CdS coating on the surface masking the diffraction peak of CdSe [27,28]. The diffraction peak of CdSe@CdS@ZnS had a red shift compared with CdSe@CdS, which was attributed to the heterogeneous epitaxial growth of zb-ZnS (JCPDS no. 05-0566) [29,30]. In summary, it could be deduced that the synthesized nanomaterials were composed of the three NCs.

### 2.2. Optical Properties of Materials

We used a UV-vis spectrophotometer and a fluorescence spectrometer to observe the changes in the optical properties of the materials. First, as shown in Figure 3a, multiple exciton peaks could be seen in the absorption spectrum of CdSe QDs, which were caused by the quantum size effects [25]. The first exciton absorption peak of CdSe QDs was located at 593 nm. When the CdS shell was grown on CdSe QDs, a red shift of the first exciton absorption peak was observed. The slight absorption spectrum of the CdSe@CdS core@shell structure at about 640 nm was attributed to the low-intensity absorption band of CdSe. However, this phenomenon was not observed in CdSe@CdS@ZnS, indicating a change in the charge-transfer mode between it. The photoluminescence (PL) spectra are shown in Figure 3b, which indicate that CdSe modified with CdS exhibited two fluorescence peaks: 509 nm for the outer layer of CdS and 641 nm for the core of CdSe. Due to the quantum size effects, the characteristic diffraction peaks were red-shifted compared to pure CdSe QDs [25]. After being modified with ZnS, the fluorescence spectrum of CdSe@CdS@ZnS still exhibited these two fluorescence peaks, but the fluorescence intensity was improved. This indicated that ZnS could effectively restrict the photogenerated electrons and holes within the interior of CdSe@CdS, thus greatly improving the luminescent properties of the materials [31].

In addition, the band gaps of CdSe, CdS and ZnS were calculated according to the UV-Vis absorption spectra (Figure 3a and Appendix A). Since CdSe, CdS and ZnS are all direct-gap semiconductors, the band gap can be determined via linear extrapolation from the absorption shoulder to (Ahν)^1/2^ to hν. According to the Tauc diagram of Figure 4a–c, the band-gap energy (*E*_g_) of CdSe, CdS and ZnS is 1.94  eV [32], 2.12  eV [33,34,35] and 3.62  eV [5], respectively. The band positions of CdSe, CdS and ZnS can be calculated using the following empirical formulae [36,37]:*E*_VB_ = *X* − *E*_c_ + 1/2*E*_g_,(1)
*E*_CB_ = *E*_VB_ − *E*_g_.(2)

In the empirical formulae, *X* is the geometric mean of the absolute electronegativity of each atom in the semiconductor [38], *E*_c_ is the energy of free electrons on the hydrogen scale (*E*_c_ = 4.5 eV), *E*_g_ is the band gap of the semiconductor, *E*_VB_ is the valence band potential, and *E*_CB_ is the conduction band potential [39]. The *X* values for CdSe, CdS and ZnS were calculated to be 5.05, 5.19 and 5.26, respectively. According to these parameters, the energy value of the valence band (VB) was calculated to be 1.52 eV, 1.75 eV and 2.57 eV, respectively, and the energy value of the conduction band (CB) was calculated to be −0.42 eV, −0.37 eV and −1.05 eV, respectively. The specific values of the band positions are shown in Appendix A. Combined with the above results, the band structure of CdSe@CdS and CdSe@CdS@ZnS is shown in Figure 4d. It could be concluded that CdSe@CdS constituted a type-II band structure. After it was modified by ZnS, CdSe@CdS@ZnS constituted a type-I band structure.

In addition, we also studied the influences of different synthetic reaction times on the optical properties of CdSe@CdS. The analysis was performed by taking equal amounts of the samples (1 mL) from the three-necked flask at different reaction times and immediately cooled them to room temperature with hexane. As shown in Figure 5a,b, the obtained samples were tested using the UV-vis absorption spectra and fluorescence spectra. We found that, with an extension of reaction time, the UV-vis absorption peaks of CdSe gradually weakened until they disappeared, and the fluorescence peaks of the CdSe phase and the CdS phase showed a red shift, indicating that the thickness of the CdS shell had gradually increased [40].

At the same time, we also examined the influence of adjusting the amount of Zn on the optical properties of CdSe@CdS@ZnS while keeping the same molar ratio of Zn to S. Figure 6a,b show the absorption and fluorescence spectra of the multicomponent heterostructures. The results show that the fluorescence peaks of the CdSe phase and CdS phase were still maintained in the multicomponent heterojunctions, indicating that the fluorescence peaks of the materials were not significantly affected by changing the amount of ZnS. The limiting effects of ZnS on the photogenic charge in CdSe@CdS were further proved.

### 2.3. Fluorescence Lifetime of Materials

To further verify the excellent optical properties of CdSe@CdS@ZnS, the fluorescence lifetime of the materials was characterized using a time-resolved transient fluorescence spectrometer. Figure 7 shows the time-resolved PL kinetic profiles of the CdSe QDs, CdSe@CdS and CdSe@CdS@ZnS. In Appendix A, the average fluorescence lifetime of single CdSe QDs is 41.4 ns, while the fluorescence quantum efficiency is 15.89%. Because CdSe @ CdS constitutes a type-II band structure, the spatial separation of photogenerated charges leads to a decrease in quantum efficiency [41]. Therefore, after CdS coating to form a CdSe @ CdS core–shell structure, its fluorescence lifetime increased to 59.0 ns, but its fluorescence quantum efficiency decreased to 6.32%. After it was modified by ZnS, CdSe@CdS@ZnS constituted a type-I band structure, and the fluorescence lifetime reached 88.8 ns, which was about 2 times higher than that of CdSe QDs and higher than the previous report (Appendix A). The quantum efficiency of CdSe@CdS@ZnS was increased to 17.05%. To further confirm that coating wide-bandgap semiconductors to form a type-I semiconductor heterojunction could enhance the luminescent quantum efficiency and extend the lifetime [42], we synthesized CdSe@CdS@ZnSe for comparison. As shown in Appendix A, the particle size of CdSe@CdS@ZnSe was 17.2 ± 0.2 nm, and the diffraction peaks of zinc-blende ZnSe (zb-ZnSe, JCPDS no. 37-1463) existed in the material (Appendix A). Moreover, the fluorescence peaks of the CdS phase and CdSe phase still appeared at corresponding positions in CdSe@CdS@ZnSe (Appendix A). As shown in Figure 7 and Appendix A, the fluorescence quantum efficiency of CdSe@CdS@ZnSe is less than 1% and the luminescence lifetime is 11.5 ns. Therefore, ZnSe and CdS also constituted a type-II band structure (Appendix A). All photogenerated electrons transfer to the conduction band of CdS, while the holes transfer to the valence bands of CdSe and ZnSe.

### 2.4. Charge-Transfer Mechanism between Heterojunctions

Based on the above results, we proposed the conversion of type II to type I in multicomponent heterojunctions, which could greatly improve the fluorescence quantum efficiency and fluorescence lifetime of the materials. As shown in Figure 8, photogenerated electrons were transferred from the conduction band of CdSe to that of CdS in CdSe@CdS, while photogenerated holes were transferred from the valence band of CdS to that of CdSe, forming a semiconductor heterojunction of type II. This was consistent with the results of the electron-and-hole separation states formed at the fluorescence peak of 641 nm [25]. After the modification of CdSe@CdS by wide-band-gap ZnS, the type-II heterojunction structure was transformed into type-I heterojunction structure, and photogenerated electrons and holes were confined inside the CdSe@CdS@ZnS, which eliminated the surface defect states, thus improving the fluorescence quantum efficiency and prolonging the fluorescence lifetime of the materials [40].

## 3. Experimental Section

### 3.1. Materials

The materials included tetramethylammonium hydroxide pentahydrate (97%+, Adamas-beta), cadmium acetate dihydrate (Cd(Ac)_2_·2H_2_O, AR, Shanghai Aladdin Biochemical Technology Co., Ltd., Shanghai, China), zinc acetate dihydrate (Zn(Ac)_2_·2H_2_O, 99.99% metals basis, Shanghai Aladdin Biochemical Technology Co., Ltd., Shanghai, China), cadmium oxide (CdO, AR, Shanghai Aladdin Biochemical Technology Co., Ltd.), stearic acid (99%, Adamas-beta), octade cylamine (80%+, Adamas-beta), selenium (Se, ≥99.999% metals basis, Shanghai Aladdin Biochemical Technology Co., Ltd.), sublimed sulfur (S, AR, Shanghai Aladdin Biochemical Technology Co., Ltd.), trioctylphosphine (TOP, 90%, Shanghai Aladdin Biochemical Technology Co., Ltd.), trioctylphosphine oxide (TOPO, 90%, Aldrich), Tri-n-Butylphosphine (TBP, 98%+, Adamas-beta), oleic acid (OAc, AR, Shanghai Aladdin Biochemical Technology Co., Ltd.), oleylamine (OAm, C18: 80–90%, Shanghai Aladdin Biochemical Technology Co., Ltd.), 1-octadecene (ODE, 90%, Sigma-Aldrich, St. Louis, MO, USA), octanoic acid (Hoc, 99%, Adamas-beta), hexane (AR, General-reagent, Belmont, NC, USA), trichloromethane (CHCl_3_, AR, Shanghai Hushi Chemical Co., Ltd., Shanghai, China), methyl alcohol (MeOH, 99.8%, Adamas-beta), and ethanol absolute (AR, General-reagent).

### 3.2. Methods

#### 3.2.1. Synthesis of the Cadmium Oleate (Cd(Ol)_2_)

First, 20 mmol (3.6246 g) of tetramethylammonium hydroxide pentahydrate and 6.4 mL of OAc were added to 50 mL of MeOH. The mixture was vigorously stirred to form homogeneous solution A. Then, homogeneous solution B was formed by adding 10 mmol (2.6653 g) of Cd(Ac)_2_·2H_2_O to 50 mL of MeOH and stirring vigorously. Then, solution B was slowly added to solution A under stirring and a milky white cadmium oleate precipitate was immediately produced. After all the mixture was added, the solution was strongly stirred for 20 min. Finally, the precipitate was centrifuged with methanol about 3 times and then dried in a vacuum oven at 40 °C. After drying, the product was stored in a vial.

#### 3.2.2. Preparation of Selenium Precursor

Se powder at 10 mmol (0.7896 g) was dissolved in 2.36 g (2.91 mL) TBP to prepare 0.1 mol/L TBP-Se solution, and then it was diluted with 6.85 g (8.68 mL) ODE to obtain selenium precursor.

#### 3.2.3. Preparation of Mixed Precursor

First, 0.3 mmol (0.2277 g) of Cd(Ol)_2_ and 0.6 mmol (0.0192 g) of S powder were added to a vial (5 mL). Subsequently, a total of 3 mL of mixed precursor solution was prepared by adding 1.5 mmol (0.5 mL) of OAm, 1.5 mmol (0.24 mL) of HOc and 2.26 mL of ODE to the above vial. The mixed solution was ultrasonically dispersed at 50 °C and stored after complete dissolution.

#### 3.2.4. Synthesis of CdSe Core Nanocrystals

In this process, 0.2 mmol (0.0256 g) of CdO and 0.8 mmol (0.2277 g) of stearic acid were added to a three-necked flask with 10 mL of ODE. The flask was then mounted on a heating magnetic stirrer. The mixture was degassed at room temperature for 30 min and then heated at 270 °C for 1 h under a nitrogen atmosphere until the liquid turned light yellow. It was cooled to room temperature, and 0.5 g of TOPO and 1.5 g of stearic amine were added to the above solution. Then, it was heated to 60 °C and stirred to dissolve the solid completely. It was degassed again until the solution had no bubbles. Then, it was pumped with N_2_ and stirred for 10 minutes, before heating to 290 °C. At this temperature, 1 mL of the TBP-Se solution was injected rapidly. After the injection, the temperature was reduced to 250 °C for 5 min and then cooled to room temperature. The precipitate was centrifuged 1–2 times with 5 mL of hexane and 15 mL of ethanol and dispersed in 5 mL of chloroform for further characterization.

#### 3.2.5. Synthesis of CdSe@CdS Core@Shell Structure

CdSe core nanocrystals in 500 μL of chloroform solution and 7.5 mL of ODE were placed in a three-necked flask. The flask was then mounted on a heating magnetic stirrer. The mixture was degassed at 120 °C for 30 min until the solution had no bubbles. Then, it was heated to 230 °C at a rate of 18 °C/min under a nitrogen atmosphere. At this temperature, 3 mL of mixed precursor started to be injected at a rate of 1.5 mL/h. At the same time, the temperature was increased to 250 °C at the same rate. When the reaction time was 10, 30, 60 and 120 min, a small sample was extracted from the mixture for testing. Finally, the solution was cooled to room temperature and centrifuged 1–2 times with 5 mL of hexane and 15 mL of ethanol. This was then dispersed in 5 mL of CHCl_3_ for further characterizations.

#### 3.2.6. Synthesis of CdSe@CdS@ZnS

An x mmol (x = 0.3, 0.6, 0.9 mmol) of sulfur powder and 2 mL of TOP were put into a 5 mL test flask for pre-ultrasonic dispersion and dissolution to obtain the precursor. Then y mmol Zn(Ac)_2_·2H_2_O (x:y = 1:2, y = 0.6, 1.2, 1.8 mmol) was added to a three-necked flask with 2 mL of OAc and 6 mL of ODE. The flask was mounted on a heating magnetic stirrer. The mixture was degassed at 120 °C for 30 min and then heated at 250 °C for 1 h under a nitrogen atmosphere to ensure complete dissolution of the solid. After that, the solution was cooled to 70 °C, and 1 mL of CdSe@CdS chloroform solution was quickly injected into the flask and degassed for 30 min, followed by heating at 250 °C under a nitrogen atmosphere. Then, the precursor was slowly injected into the above solution at a rate of 0.1 mL/min and reacted for another 40 min after the injection processes. The solution was cooled to room temperature and centrifuged 1–2 times with 5 mL of hexane and 15 mL of ethanol. Finally, this was dispersed in 5 mL of CHCl_3_ for further characterizations.

#### 3.2.7. Characterizations

The morphological structures of the samples were measured using scanning electron microscopy (FESEM, Hitachi S4800), transmission electron microscopy (TEM, Hitachi HT7820) and high-resolution transmission electron microscopy (HRTEM, Philips CM100). To determine the crystal phases, X-ray diffraction (XRD) patterns were obtained using a Rigaku Dmax-3C with Cu Kα irradiation (λ = 1.5406 Å). The UV-vis absorption spectra were acquired using a Shimadzu 1900i spectrophotometer. Steady-state fluorescence spectroscopic measurements were performed using a fluorescence spectrophotometer (Hitachi F-7000). Fluorescence quantum efficiency and fluorescence lifetime were measured using a time-resolved transient absorption spectrometer and a spectrometer with an integrating sphere (Edinburgh Instrument, EI FLS1000).

## 4. Conclusions

In this study, CdSe@CdS@ZnS nanocrystals with multi-component heterostructures were constructed utilizing the band structures between semiconductors to transition from a type-II to a type-I structure, resulting in improved luminescence quantum efficiency and increased fluorescence lifetime. Compared with CdSe quantum dots, the luminescence quantum efficiency of the CdSe@CdS core@shell structure of the type-II heterojunction decreased by 60%, while the fluorescence lifetime increased by 42.5%. The luminescence quantum efficiency of the type-I heterojunction CdSe@CdS@ZnS was 17.05%, and the fluorescence lifetime was 88.8 ns. Compared with nanocrystalline CdSe QDs, its fluorescence lifetime was increased by about 2 times. This study provided a new method and concept for future research on the construction and charge transfer of multicomponent semiconductor nanocrystals for enhanced PL properties.

## Figures and Tables

**Figure 1 materials-16-07007-f001:**
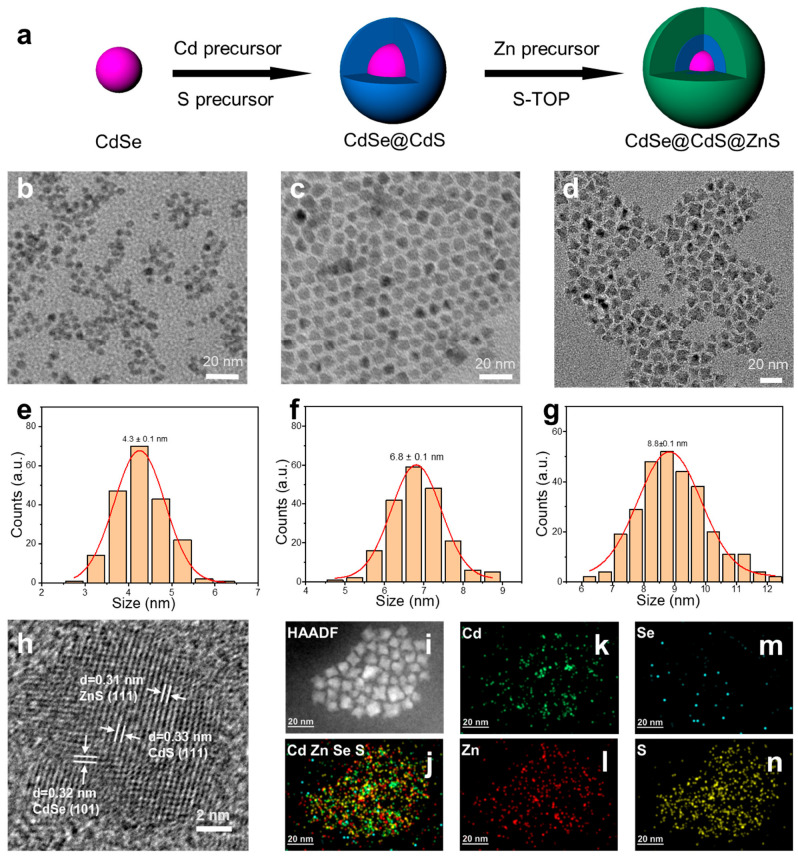
(**a**) The scheme of the synthetic processes of CdSe@CdS@ZnS. Representative TEM images of (**b**) CdSe QDs; (**c**) CdSe@CdS; and (**d**) CdSe@CdS@ZnS. Size distribution diagrams of (**e**) CdSe QDs; (**f**) CdSe@CdS; and (**g**) CdSe@CdS@ZnS. (**h**) HRTEM image. (**i**–**n**) HAADF-STEM-EDS elemental mapping images of CdSe@CdS@ZnS.

**Figure 2 materials-16-07007-f002:**
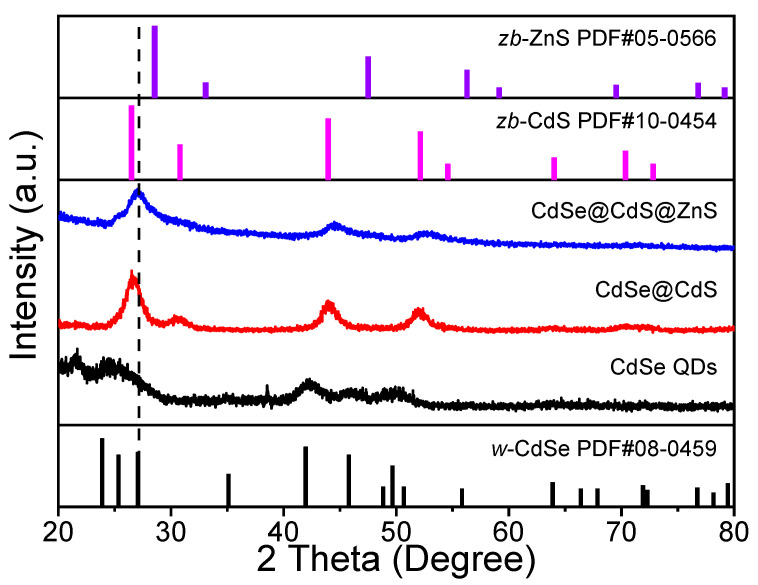
XRD patterns of CdSe QDs, CdSe@CdS and CdSe@CdS@ZnS.

**Figure 3 materials-16-07007-f003:**
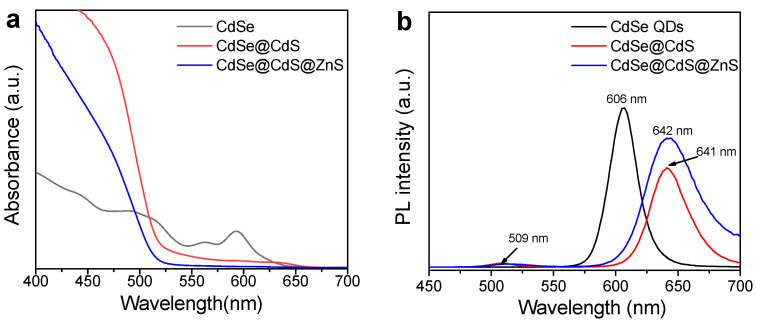
UV-vis absorbance spectra (**a**) and photoluminescence (PL) spectra (**b**) of CdSe QDs, CdSe@CdS and CdSe@CdS@ZnS.

**Figure 4 materials-16-07007-f004:**
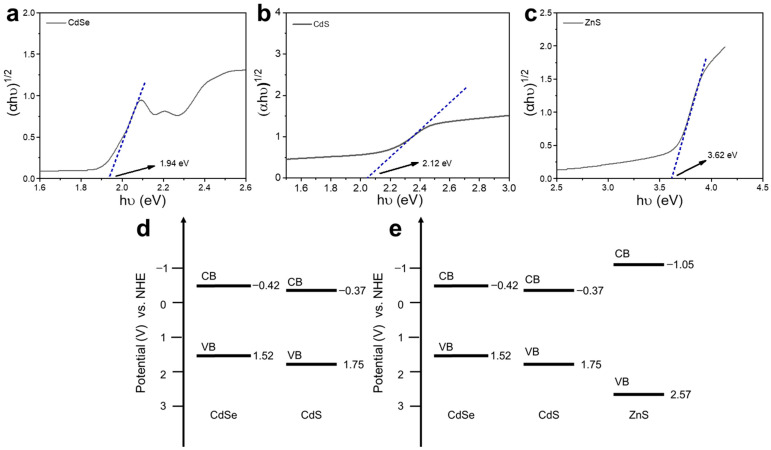
Tauc plots to estimate bandgaps of (**a**) CdSe QDs, (**b**) CdS NPs and (**c**) ZnS NPs. The band structure of (**d**) CdSe@CdS and (**e**) CdSe@CdS@ZnS.

**Figure 5 materials-16-07007-f005:**
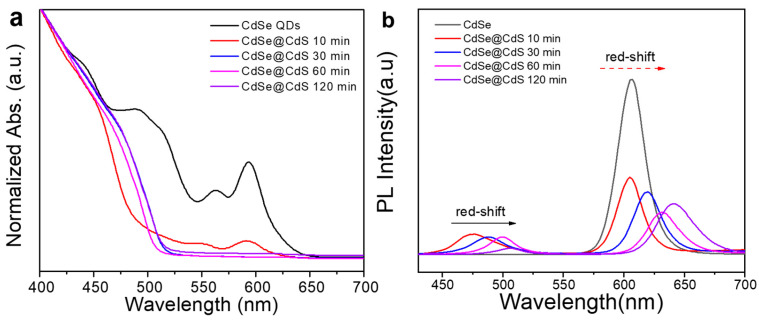
UV-vis absorbance spectra (**a**) and photoluminescence (PL) spectra (**b**) of CdSe@CdS at different synthetic reaction times.

**Figure 6 materials-16-07007-f006:**
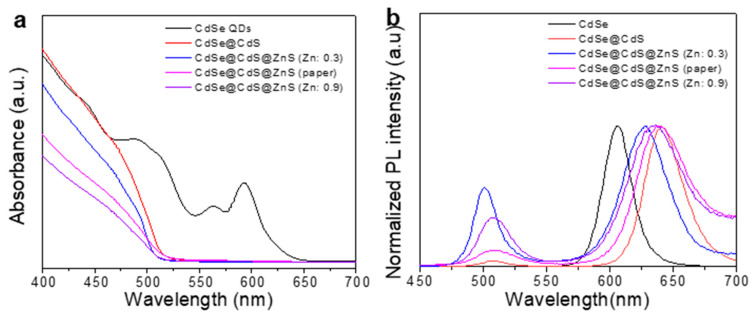
UV-vis absorbance spectra (**a**) and photoluminescence (PL) spectra (**b**) of CdSe@CdS@ZnS with different amounts of Zn.

**Figure 7 materials-16-07007-f007:**
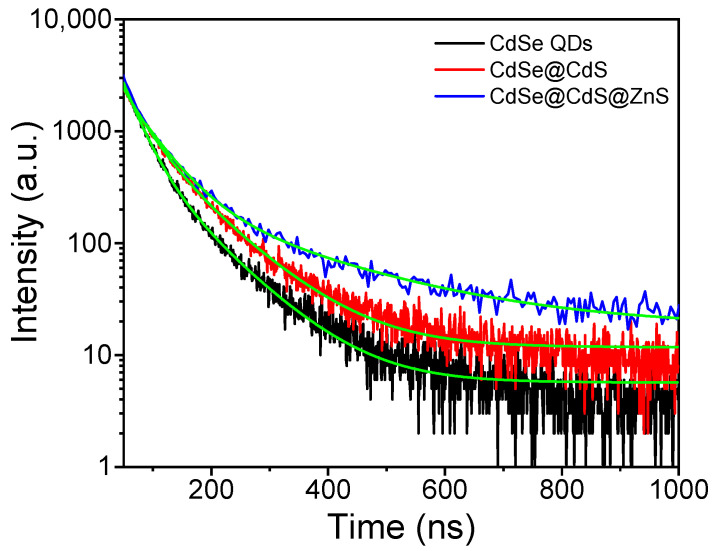
Time-resolved PL at the main peak for CdSe QDs, CdSe@CdS and CdSe@CdS@ZnS. Best-fit lines obtained using biexponential fitting are shown in green.

**Figure 8 materials-16-07007-f008:**
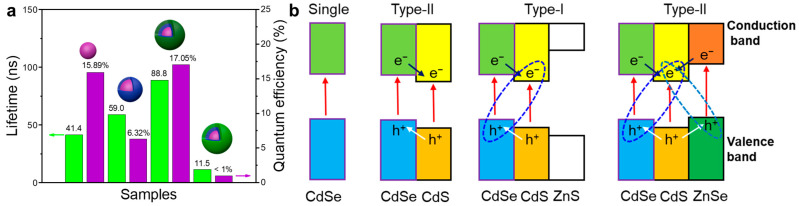
(**a**) The PL lifetime and quantum efficiency of CdSe QDs, CdSe@CdS, CdSe@CdS@ZnS and CdSe@CdS@ZnSe. (**b**) The energetic band positions of all the samples as comparisons, and photoinduced carrier transfer in the type-II and type-I band alignments.

## Data Availability

The data presented in this study are available from the corresponding author upon request.

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
