# Peer review of "Type-I CdSe@CdS@ZnS Heterostructured Nanocrystals with Long Fluorescence Lifetime"

_materials, 2023, doi:10.3390/ma16217007_

Round 1

Reviewer 1 Report

Comments and Suggestions for Authors

The manuscript (MS) under review that entitled “Type-I CdSe@CdS@ZnS heterostructured nanocrystals (NC) exhibit high photoluminescence quantum efficiency and long lifetime” deals with preparation and characterization of CdSe@CdS@ZnS nanostructures. The results reported in the MS fall in scope of “Optical and Photonic Materials” section of the journal “Materials”. At the same time, the MS in its’ current form has many issues those must be solved before publication. Thus, I recommend major revision.

 Below some comments and suggestions that could help to improve MS.

1)  It is hard to understand what a novelty of the study. The discussion on transformation from type II to type I heterojunctions and its effect on luminescence properties is speculative. Please state clear in the manuscript (“Introduction” or somewhere else) what new results were obtained.

2) It is supposed that changes in luminescence properties of the studied NC are related with transformation of the CdSe@CdS@ZnS from type II to type I state due to covering of CdSe@CdS nanocrystals with ZnS shell. However, in the supplementary, authors include the results for larger NC ( 17 nm). For such larger NC authors found quantum efficiency near 1% only. It is unlikely, that the CdSe@CdS@ZnS system has transformation from type II heterojunction to type I and then back to type II with increasing of thickness of the ZnS shell. More likely, the ZnS shell play “protective” role for CdSe@CdS decreasing number of defects those act as charge traps. Actually, similar assumption can be found in the MS at lines 207-211.

3) I recommend to change MS title to “The CdSe@CdS@ZnS heterostructured nanocrystals with long photoluminescence lifetime”. The reported in the MS luminescence quantum efficiency is 17.05%, that is not so high as for similar system (up to 84.7% - Kim, J., Hwang, D. W., Jung, H. S., Kim, K. W., Pham, X. H., Lee, S. H., ... & Jun, B. H. (2022). High-quantum yield alloy-typed core/shell CdSeZnS/ZnS quantum dots for bio-applications. Journal of nanobiotechnology20(1), 22.; or near 80 % - Talapin, D. V., Mekis, I., Götzinger, S., Kornowski, A., Benson, O., & Weller, H. (2004). CdSe/CdS/ZnS and CdSe/ZnSe/ZnS Core− Shell− Shell Nanocrystals. The Journal of Physical Chemistry B108(49), 18826-18831.; 50% - Grabolle, M., Ziegler, J., Merkulov, A., Nann, T., & ReschGenger, U. (2008). Stability and fluorescence quantum yield of CdSe–ZnS quantum dots—influence of the thickness of the ZnS shell. Annals of the New York Academy of Sciences1130(1), 235-241.)

4) Lines 19-21. “The results show that the charge transfer between semiconductor heterojunctions of CdSe@CdS@ZnS resulting in the band transformation from type-II to type-I leads to the increase of photoluminescence lifetime from 41.4 ns to 88.8 ns compared with the pristine CdSe QDs, and the photoluminescence quantum efficiency reaches 17.05%.” – The results only show that covering with ZnS shell led to improvement of luminescence properties of CdSe@CdS heterostructure, but not charge transfer.

5)    Lines 23-24. “This paper provides the fundamental study and a new route for studying light-emitting devices and biological imaging based on multicomponent QDs.”  - Currently, it is not clear where “fundamental study” and where “new route for studying….” mentioned by authors. The CdSe@CdS@ZnS systems were obtained and studied earlier by the similar methods (1 - Wang, Y., Ta, V. D., Gao, Y., He, T. C., Chen, R., Mutlugun, E., ... & Sun, H. D. (2014). Stimulated emission and lasing from CdSe/CdS/ZnS coremultishell quantum dots by simultaneous threephoton absorption. Advanced Materials26(18), 2954-2961., 2 - Reiss, P., Protiere, M., & Li, L. (2009). Core/shell semiconductor nanocrystals. Small5(2), 154-168., 3- Manna, L., Scher, E. C., Li, L. S., & Alivisatos, A. P. (2002). Epitaxial growth and photochemical annealing of graded CdS/ZnS shells on colloidal CdSe nanorods. Journal of the American Chemical Society124(24), 7136-7145).

6)    Lines 29-30. “…colloidal nanocrystals have aroused extensive attention in the field of basic research and practical applications due to their excellent optical and photoelectrical properties [1-7]” – All the seven cited references are the papers from the same scientific group. There are a lot of other appropriate studies, that can be cited. Moreover, some of the ref. 1-7 are related with too different type of QD than studied in the MS under review (like Co-Pt alloy in ref 4 or metal-organic frameworks in refs 1 and 7).

7)    Lines 50-51. “…there are few reports on the construction of multi-component structures from type-II to type-I to enhance the luminescence properties of the material.” – please provide references.

8)    Lines 65-66. “… quantum efficiency increased from 6.32% to 17.05%, showing excellent luminescence properties.” – It is good but not excellent properties (see comment 3).

9)    Line 69. “We synthesized a novel CdSe@CdS@ZnS heterojunction nanocrystals…” – what novelty of this NC in comparison with structures those are reported in literature earlier (e.g. 1- Jones, M., Lo, S. S., & Scholes, G. D. (2009). Quantitative modeling of the role of surface traps in CdSe/CdS/ZnS nanocrystal photoluminescence decay dynamics. Proceedings of the National Academy of Sciences106(9), 3011-3016. 2- Wang, Y., Ta, V. D., Gao, Y., He, T. C., Chen, R., Mutlugun, E., ... & Sun, H. D. (2014). Stimulated emission and lasing from CdSe/CdS/ZnS coremultishell quantum dots by simultaneous threephoton absorption. Advanced Materials26(18), 2954-2961. 3 - Liu, Y., Dai, F., Zhao, R., Huai, X., Han, J., & Wang, L. (2019). Aqueous synthesis of core/shell/shell CdSe/CdS/ZnS quantum dots for photocatalytic hydrogen generation. Journal of Materials Science54(11), 8571-8580.)

10)                  Description of synthesis is very confusing. There are 2 subsections in MS and one in supplementary that describing obtaining of nanocrystals. I suggest making only one description in the 2.1 section of the manuscript: line 72 the sentence: selenium precursor as the selenium source has no sense. The name of the selenium-containing substance should be included. Line 76 “injecting Cd and S mixed precursors solution” – the substances have to be included in the synthesis description.

11) Line 90. X-Ray powder diffraction patterns are used not for morphology characterization, but for crystal phase identification. This sentence has to be changed.

12) Lines 95-96. Red shift of the XRD peaks, likely, related with overlapping of peaks from all the three phases of NC. Using "red-shift" in case of scattering is doubtful. 

13)  It is better to provide separate images for each chemical element at HAADF-STEM (Fig 1j). From current image it is hard to see that studied CdSe@CdS@ZnS are core/shell nanostructures but not separate CdSe, CdS and ZnS nanocrystals.

14)  Lines 112-114. “The slight absorption spectrum of the CdSe@CdS core@shell structure at about 640 nm was attributed to the delocalization of electrons and holes after the formation of heterojunction, resulting in the charge separation state between CdS and CdSe[30].”  - What is slight absorption spectrum means? Maybe absorption band of low intensity?

15)  Lines 119-120. “Due to the quantum size effect, the characteristic diffraction peaks were red-shifted compared to pure CdSe QDs [25].” – Diffraction peaks red-shifted? It is nonsense. Probably authors mean luminescence band maximum?

16)  Line 132. What a reason for references near Eg values if these values were obtained by authors from their results? Does the Eg obtained (1) separately for CdSe, CdS and ZnS or (2) for the studied nanocrystals? In the case (1) such analysis has not much sense, because a mismatch between CdSe, CdS and ZnS crystal lattices must affect the energy structure of heterojunctions.

17)  Lines 139-142 and Fig. 4.d,e. Please use either potentials or energies. According to ref 36, the formulas 1 and 2 are for potentials but not energies.

18)  Line 151. “…small amount of liquid…” – what liquid was used? Please provide information what samples were studied by luminescence spectroscopy (suspensions or powders).

19)      Figure 8 has wrong captions.

20)   Line 305-307. It is hard to agree with the sentence “This study provided a new method and concept for future research on the construction and charge transfer of multicomponent semiconductor nanocrystals for enhanced PL properties.” What method is new? Synthesis of nanocrystals (the similar methods can be found in literature)? Using of Eg determination for revealing of heterojunction type (it looks like that authors have been used wrong NCs for Eg determination)? Please clarify somewhere (Introduction or Discussion) in the MS what is new in the present study.

Reviewer 2 Report

Comments and Suggestions for Authors

The authors prepare CdSe@CdS@ZnS core@shell heterojunction to enhance charge transfer between semiconductor heterojunctions and increase of photoluminescence lifetime. The article is an interesting study in the field of quantum dots applications. The manuscript is well structured, and the results are presented in relation to the literature studies.

In my opinion, the manuscript could be considered for publication after careful revision

1.      Authors should evident the thickness of  CdS and ZnS of the CdSe@CdS@ZnS core@shell heterojunction.

2.      Authors should discuss (in advanced) and compare the XRD pattern of  CdSe@CdS and CdSe@CdS@ZnS samples as shown in Fig. 2

3.      To understand bandgaps of pristine CdSe QDs, CdS NPs, and ZnS NPs. The method to prepare such NPs samples should be provided in the support information. And the morphology of such samples should be provided.

4. I highly recommend the authors to compare their results with previous studies.

Reviewer 3 Report

Comments and Suggestions for Authors

I have a few comments regarding to this manuscript.

The CdSe@CdS@ZnS have an average diameter ~12.7nm, figure 1h in the HRTEM it looks like the diameter is much smaller than the 12.7 nm. In addition to the lattice spacing they claim the presence of the CdSe, CdS, and ZnS, but it looks like all the three values are within the same crystal with no apparent difference in the d value. In addition, I can see other crystal lattices, but they are not labeled. 

Figure 1j does not prove the presence of CdSe@CdS@ZnS, it only proves the presence of the elements Cs, Se, S, Zn. If you want to use this technique it should be an EDX line scan for a single CdSe@CdS@ZnS. With this data, it looks more like an alloy rather than a three-shell structure. 

Why the XRD peaks are sharper for CdSe@CdS than for CdSe@CdS@ZnS.

Reviewer 4 Report

Comments and Suggestions for Authors

The paper by Wang et al. is thorough study of new type of heterostructures. Authors describe synthesis and physical characterisation of a new material. In my opinion, the study might be interesting to the readers of the journal Matherials, and should be accepted after minor corrections.

First, the introduction or discussion should be broadened by short description of the properties of other similar materials. 

I lack also the information about reproductability of synthesis. How are the properties of nanostructure preserved in second/third repetition?

Additional small point - the title should be "... long fluorescence lifetime", not "long lifetime".

In the abstract, authors also mention about application of their nanostructures for biological imaging, while not providing any data about matherial stability in typical "biological imaging" conditions. This point is also not discussed thoroughly, what should be changed. 

Round 2

Reviewer 1 Report

Comments and Suggestions for Authors

The authors have provided satisfactory answers to all the questions and therefore, the revised manuscript can be recommended for publication.

Reviewer 3 Report

Comments and Suggestions for Authors

The manuscript is good in the present form.
